# Incivility Is Associated with Burnout and Reduced Compassion Satisfaction: A Mixed-Method Study to Identify Causes of Burnout among Oncology Clinical Research Coordinators

**DOI:** 10.3390/ijerph182211855

**Published:** 2021-11-12

**Authors:** Jennifer S. Mascaro, Patricia K. Palmer, Marcia J. Ash, Caroline Peacock, Cam Escoffery, George Grant, Charles L. Raison

**Affiliations:** 1Department of Family and Preventive Medicine, School of Medicine, Emory University, Atlanta, GA 30322, USA; 2Woodruff Health Sciences Center, Department of Spiritual Health, Emory University, Atlanta, GA 30322, USA; kpalmer01@gmail.com (P.K.P.); caroline.peacock@emory.edu (C.P.); ghgrant@emory.edu (G.G.); raison@wisc.edu (C.L.R.); 3Department of Behavioral, Social and Health Education Sciences, Rollins School of Public Health, Emory University, Atlanta, GA 30322, USA; marcia.j.ash@emory.edu (M.J.A.); cescoff@emory.edu (C.E.); 4Winship Cancer Institute, Emory University, Atlanta, GA 30322, USA; 5School of Human Ecology, University of Wisconsin-Madison, Madison, WI 53706, USA

**Keywords:** clinical research coordinators, clinical trials, burnout, incivility

## Abstract

While oncology clinical research coordinators (CRCs) experience a combination of factors that are thought to put them at increased risk for burnout, very little research has been conducted to understand the risk factors associated with burnout among CRCs. We used a mixed-method approach, including self-report questionnaires to assess burnout and compassion satisfaction, as well as individual and interpersonal variables hypothesized to impact CRC well-being. We also conducted a focus group to gain a more nuanced understanding of coordinators’ experiences around burnout, teamwork, resilience, and incivility. Coordinators reported relatively moderate levels of burnout and compassion satisfaction. Resilience, sleep dysfunction, stress, and incivility experienced from patients/family were significant predictors of burnout. Resilience and incivility from patients/family were significant predictors of compassion satisfaction. Themes that emerged from the focus group included that burnout is triggered by feeling overwhelmed from the workload, which is buffered by what was described as a supportive work culture based in teamwork. This study identified variables at the individual and interpersonal level that are associated with burnout and compassion satisfaction among oncology CRCs. Addressing these variables is of critical importance given that oncology CRCs and team-based coordinator care are vital to the success of clinical trials.

## 1. Introduction

Clinical research coordinators (CRCs) play a vital role in clinical research activities and interactions with clinical study participants, and they are integral to the success of the clinical trials upon which most oncology advances depend [1,2]. However, CRCs report high levels of emotional exhaustion and similar or worse levels of burnout [3], compared to other health care professionals experiencing a well-documented epidemic of burnout [4,5,6,7,8,9,10,11]. Despite these high rates of suffering among CRCs and their importance to clinical research, little research has been conducted to understand the risk factors associated with CRC burnout.

Previous research indicates that burnout is driven by both individual and organizational factors, including excessive workload and work-related stress, inefficiencies, interpersonal and moral conflicts, and a clinical and institutional climate that limits the advancement, autonomy, and flexibility of employees [8,11,12]. Although little research has been conducted to understand burnout among CRCs, coordinators experience a combination of factors that are thought to put them at increased risk for burnout. CRCs work within interprofessional teams that are highly hierarchical, in which coordinators garner lower levels of compensation, authority, and status compared to the physicians with whom they interact [13,14]. Coordinators often work in multidisciplinary and multi-team systems, both of which present challenges to communication, cohesion, and coordination [15]. Other research indicates that coordinators often experience distress and anxiety over gaps in available resources that can jeopardize trial success or patient outcomes [16]. The risks for oncology CRCs are likely elevated further when compared with their CRC peers in other areas of health care. Coordinators working in oncology departments help manage patients who have high levels of pain, help implement treatment plans with adverse side-effect profiles, and experience a relatively higher frequency of patient death [17,18,19].

While the prevalence of burnout among CRCs is concerning, extensive research also highlights protective factors that are important to consider in understanding burnout within the clinical research team. For example, the presence of positive social support among colleagues reduces burnout and improves both job performance [20] and attitudes [21]. In fact, positive workplace relationships moderate the relationship between stressors and burnout, in effect buffering health care employees from the harmful effects of an elevated workload [22]. Factors that are protective against burnout also positively contribute to compassion satisfaction, defined as the ability to derive pleasure and gratification from caregiving [23,24]. Although burnout and compassion satisfaction are thought to be related [24], the correlation is often weak, and it is clear that they are not simply inverse constructs [25].

The overall goal of this study was to evaluate the individual and interpersonal factors associated with burnout and compassion satisfaction among clinical research coordinators working in a National Cancer Institute (NCI)-designated Comprehensive Cancer Center. Existing frameworks posit that effective approaches to reducing burnout can target multiple levels, including the individual level, the social and team level, and the organizational level [26]. For this reason, we evaluated the relationship between burnout and compassion satisfaction and factors at both the individual and interpersonal levels that are hypothesized to impact health care employee well-being. Regarding the former, we measured factors shown in previous research to be important to the well-being of health care employees, including stress, resilience, and depression [27]. Regarding interpersonal factors important to well-being, we were interested in CRCs experiences of incivility from their colleagues, leadership, and from patients, given the well-documented negative effects of incivility on morale and performance [28,29]. We used a mixed-method approach, including self-report questionnaires and a focus group to gain a more nuanced understanding of coordinators’ experiences around burnout, teamwork, resilience, and incivility.

## 2. Methods

### 2.1. Study Overview

The research was conducted with our university’s Institutional Review Board’s approval as part of a larger randomized controlled trial evaluating the effect of a team-based intervention to improve resilience and well-being among CRCs (NCT04060901). All work was carried out in accordance with the Declaration of Helsinki. Participants in the randomized controlled trial gave informed consent.

### 2.2. Participants

CRC participants work in disease-specific teams within an NCI-designated Comprehensive Cancer Center in the Southeastern United States. All coordinators (*n* = 130) were invited to participate.

### 2.3. Self-Report Measures

We administered a self-report survey battery, delivered in person or via email link (according to the CRC team preference) prior to randomization in the aforementioned clinical trial (all Cronbach’s alpha values listed below were generated from this study). To assess burnout and compassion satisfaction, we administered the Professional Quality of Life scale (ProQOL) Version 5 in English [24]. This 30 item-scale is scored on a 5-point scale (never to very often) and includes subscales for compassion satisfaction, burnout, and secondary trauma stress (STS). We did not evaluate factors associated with STS in this study as STS is a measure of the extent to which one experiences secondary exposure to traumatically stressful events, whichis less influenced by individual and interpersonal factors associated with the work environment and more related to the nature of the work itself. (Cronbach’s alpha: burnout: 0.93; compassion satisfaction: 0.93).

To assess individual factors associated with burnout, we administered:Depression, Anxiety, and Stress Scale 21 (DASS-21) [30]: The DASS-21 contains 21 items, scored on a four-point scale ranging from did not apply to me at all to applied to me very much or most of the time. DASS includes subscales for depression, anxiety, and stress. (Cronbach’s alpha: depression: 0.91; anxiety: 0.77; stress: 0.86)Connor–Davidson Resilience Scale (CD-RISC-25) [31]: The CD-RISC is a 25 item-scale, scored on a 5-point scale (not true at all to true nearly all of the time). (Cronbach’s alpha: 0.90)PROMIS Sleep Disturbance Instrument [32]: A nine-item scale that assesses self-reported perceptions of sleep quality, sleep depth, and restoration associated with sleep. It includes perceived difficulties and concerns with getting to sleep or staying asleep, as well as perceptions of the adequacy of and satisfaction with sleep (Cronbach’s alpha: 0.64).

To assess the interpersonal factors associated with burnout, we administered the Nursing Incivility Scale [33]. Respondents report on their experience of seven types of incivility coming from three sources: (1) physicians and other hospital colleagues, (2) members of their CRC team, and (3) patients and their family. Responses were summed to calculate a score for each source of incivility (Cronbach’s alpha: physicians and other hospital colleagues: 0.89; members of their CRC team: 0.91; patients and their family: 0.89).

### 2.4. Qualitative

We also conducted a focus group discussion to gain a more nuanced understanding of coordinators’ experiences around burnout, teamwork, resilience, and incivility. The focus group lasted approximately one hour and took place in a private room at the cancer center. It was conducted as part of a quality improvement initiative (institutional review board approval was waived for this component). The focus group was facilitated by a trained member of the research team and followed a semi-structured discussion guide based on the Consolidated Framework for Implementation Research (CFIR) [34], a widely used implementation science framework. Topics for discussion included wellness and resilience, work culture, and recommendations for addressing burnout and resilience. The focus group session was audio recorded and transcribed verbatim.

A deductive codebook was developed based on the CFIR guide and included pertinent constructs such as Needs and Resources, Network and Communication, Culture, Leadership Engagement, and Organizational Incentives and Rewards. Principles of thematic analysis were applied to gain a more in-depth understanding of participants’ reported experiences [35]. In addition, to identify inductive codes, the team used an open coding approach where transcripts were reviewed for recurring emergent concepts pertinent to the research question. Each code was assigned a definition and inclusion/exclusion criteria. The transcript was coded by one member of the study team and then reviewed by a second member of the team. Any discrepancies were discussed between team members until agreement was met. All coding was completed using NVivo 12.

### 2.5. Statistical Analysis

Missing items in the psychometric scales were estimated with expectation maximization [36] using other items within the scale as predictor variables. Missing items never accounted for more than 5% of total data; all items were missing at random as determined by null Little’s missing completely at random (MCAR) tests. Descriptive statistics (mean, standard deviation, and standard error for continuous variables, frequency, and % for categorical) were used to characterize CRC demographics and survey responses. All data were evaluated for normality using the Shapiro–Wilk test, and appropriate procedures were performed in cases where normality was violated. To examine the correlation between the dependent and independent variables of interest, we conducted Spearman rank-order correlations. Next, we conducted stepwise linear regression using a backward elimination method to estimate the effect of individual factors (sleep, depression, anxiety, stress, and resilience) and interpersonal factors (incivility from hospital colleagues, other coordinators on their team, and patients and family) on two dependent variables: burnout and compassion satisfaction. For each dependent variable, backward elimination was conducted by entering all independent individual variables into the model. We generated a reduced model by eliminating variables that did not significantly contribute (*p* > 0.05) to burnout/compassion satisfaction. Next, we performed the same process with interpersonal variables. Finally, in a third step to characterize the amount of variance in the dependent variables explained by both individual and interpersonal variables, we entered all individual and interpersonal variables remaining from the first two reduced (individual and interpersonal) models. If needed, non-significant predictors were removed in a last elimination step to produce a final combined model. All analyses were performed using SPSS (version 27.0 for Windows, SPSS, Inc., Chicago, IL, USA)

## 3. Results

Of the approximately 130 CRCs working in the center, 45 CRCs completed the self-report survey (Table 1). Of the CRCs enrolled in the study, 84% were female; 34% were white, 43% were African American/Black, 14% were Asian, and 9% were “other” (Mexican, multi-ethnic, multiracial, and mixed unknown); and 11% were Hispanic or Latino, 84% not Hispanic or Latino, and 5% unknown.

As shown in Table 2, average baseline levels of burnout and compassionate satisfaction were mid-range [37,38], while depression, anxiety, and stress were low [39]. Resilience was moderate, falling lower than the average scores found in some previous studies [31] and higher than was found in others [40]. Workplace incivility was in the low-to-mid range for each type of incivility, with highest relative incivility scores on the gossip/rumor and inconsiderate behavior subscales and lowest relative scores on the lack of respect and inappropriate jokes subscales. Correlation analyses indicate a significant inverse correlation between burnout and compassion satisfaction (rs = −0.62, *p* < 0.001). Burnout was significantly positively correlated with stress (rs = 0.48, *p* = 0.001), anxiety (rs = 0.53, *p* < 0.001), depression (rs = 0.54, *p* < 0.001), sleep disturbance (rs = 0.54, *p* < 0.001), and incivility experienced from patients and their family (rs = 0.38, *p* = 0.014), and negatively correlated with resilience (rs = −0.61, *p* < 0.001). Compassion satisfaction was negatively correlated with stress (rs = −0.32, *p* = 0.039), depression (rs = −0.50, *p* = 0.001), and incivility from patients (rs = −0.50, *p* = 0.001) and positively correlated with resilience (rs = 0.67, *p* < 0.001).

### 3.1. Burnout

Results of the multiple linear regression analyses using burnout as the dependent variable indicated a significant collective effect of the individual variables (depression, anxiety, stress, sleep, and resilience) on burnout (F(5, 35) = 10.72, *p* < 0.001, R2 = 0.61) (Table 3). However, neither anxiety nor depression were significant in predicting variance in burnout and were thus eliminated from the model. The reduced model for personal variables and burnout included sleep, stress, and resilience (F(3, 37) = 18.79, *p* < 0.001, R2 = 0.60). Regarding the association between interpersonal variables and burnout, incivility variables were significantly associated with burnout (F(3, 38) = 3.32, *p* = 0.03, R2 = 0.21). However, only incivility experienced from patients was significant in the reduced interpersonal model (F(1, 40) = 8.84, *p* = 0.005, R2 = 0.18). The final combined (personal and interpersonal model) included significant associations with sleep, stress, resilience, and incivility experienced from patients and their family (F(4, 36) = 17.56, *p* < 0.001, R2 = 0.66).

### 3.2. Compassion Satisfaction

Results of multiple linear regression analyses using compassion satisfaction as the dependent variable indicated that there was a significant collective effect of the personal variables (depression, anxiety, stress, sleep, and resilience) on compassion satisfaction (F(5, 35) = 6.65, *p* < 0.001, R2 = 0.49) (Table 4). However, among these variables, only resilience was significant in predicting variance in compassion satisfaction in the reduced personal model (F(1, 40) = 35.88, *p* < 0.001, R2 = 0.47). Regarding the association between interpersonal variables and burnout, incivility variables were significantly associated with burnout (F(3, 38) = 7.76, *p* < 0.001, R2 = 0.38). However, only incivility experienced from other coordinators and from patients was significant in the reduced interpersonal model (F(2, 39) = 11.79, *p* < 0.001, R2 = 0.38). When the significant individual and interpersonal variables were entered, incivility from co-workers was no longer significant; the final model included resilience and incivility experienced from patients and their family (F(2, 39) = 25.57, *p* < 0.001, R2 = 0.57).

### 3.3. Qualitative Focus Group

Five CRCs from two teams participated in a follow-up focus group session; 60% of participants were male with a mean age of 31. Emergent themes included that burnout is triggered by feeling overwhelmed from too many work responsibilities and that it is exacerbated by a lack of understanding from physicians and leadership. While CRCs described a desire for more appreciation from leadership, they stated that there is a supportive work culture among CRCs based in teamwork. Finally, they indicated that interventions to support CRC well-being and cope with burnout would be most beneficial for CRCs who are new to the role. Summary of emergent themes and representative quotes is found in Table 5.

## 4. Discussion

This mixed-method study was conducted to help redress the limited research examining burnout among CRCs. We found that both personal and interpersonal factors are associated with burnout and compassion satisfaction among coordinators working in disease-specific teams within a comprehensive cancer center. The quantitative data indicated that sleep disturbance, stress, and incivility experienced from patients and their family members are risk factors for burnout. Incivility from patients and their family was also associated with reductions in compassion satisfaction. Among personal factors, self-reported resilience was associated with enhanced compassion satisfaction and reduced burnout. These individual and interpersonal factors are highly predictive of well-being in our sample, explaining 66% of the variance in burnout and 57% of the variance in compassion satisfaction. Qualitative focus groups reinforced these findings to some extent, as coordinators reported that burnout is driven by feeling overwhelmed by work responsibilities. CRCs described high levels of team cohesion and indicated that the supportive culture among their team helps to reduce the stress from excessive work.

Previous research consistently finds a relationship between burnout and symptoms of depression and stress [27,41], and we found that relationship within our data as well. Interestingly, however, our regression analyses did not indicate that depression accounted for significant variance in burnout or compassion satisfaction. Recent qualitative research suggests that there are nuanced differences between depression and burnout, both in terms of their causes and lived experience [42]. These data are consistent with that idea and suggest that, although there is an association between burnout and depression in CRCs, there are important differences between the two. In contrast, sleep disturbance was both strongly associated with and explained significant variance in burnout, replicating a relationship that has been found in other studies of burnout risks for health care employees [43,44]. Taken together with previous studies, these data indicate that interventions or other approaches to improve sleep hygiene and reduce sleep dysfunction are a promising way to mitigate burnout.

Our study also identified interpersonal risk factors associated with burnout and compassion satisfaction. Clinical research coordinators are critical but under-studied members of interprofessional, team-based oncological care and academic medicine. They play an essential role in multiple distinct and vital aspects of clinical trials research, including participant recruitment, screening, and enrollment. They implement research protocols, interface with regulatory oversight bodies, and support the safety of clinical research participants. They also serve as the primary liaison between the health care team and patients and their family members [45]. At times, CRCs’ roles in advocating for patients, patients-turned-research-participants, and research may conflict, and coordinators must balance these roles within a large interprofessional team [46]. Previous research indicates that health care team relationships are critical to patient outcomes, and team relationship quality predicts patient mortality rates [47], treatment adherence [48], and patient safety [49]. Our data indicate that coordinators generally find support among their CRC colleagues. While incivility among coordinators was unexpectedly positively associated with compassion satisfaction, this relationship was no longer significant in the final model. Overall, coordinator civility was unrelated to burnout levels and compassion satisfaction. In the focus group, coordinators reported high levels of support among their team-members and that this is an important contributor to their well-being.

Interestingly, there were some domains in which the quantitative and qualitative findings were inconsistent. While coordinators in the focus group indicated that burnout is exacerbated by a lack of understanding or appreciation from physicians and leadership, the quantitative data did not indicate that incivility experienced from physicians and other hospital personnel was associated with burnout. Moreover, discussion of incivility from patients and their family did not arise during the focus group. It may be that the experience of incivility from patients is not as salient to coordinators, or that CRCs do not feel comfortable admitting that it occurs or that they find it distressing. A previous qualitative study of CRCs found that altruism plays a prominent role in coordinators’ orientation to their roles as well as in their day-to-day function in the interprofessional team [13]. Moreover, the authors of this study proposed that there is a gendered component to coordinator altruism, since coordinators are primarily women. Our sample was also primarily women, although the focus group had a more even gender split. It is possible that coordinators do not feel it is socially acceptable to give voice to patients’ incivility towards them or the distress it causes. Regardless, our data indicate that it is a potent factor in both burnout and compassion satisfaction. Organizational policies to improve the ways that CRCs are treated by patients and providing CRCs with tools to manage patient and family incivility will be important in reducing their burnout levels. Given that burnout is a primary motivation cited by CRCs for leaving the job [45], and CRC turnover is an ongoing problem in academic cancer centers [2], addressing this factor has the potential to be critical.

A limitation of this cross-sectional study is that we cannot definitively determine the causal relationship between the associations identified here. While we have generally interpreted the findings as indicating that sleep disruption, stress, and incivility experienced from patients impact burnout levels, it is likely that burnout is also influencing these personal and interpersonal variables. In fact, it is likely that the relationships are bidirectional, which would indicate a cycle by which problematic personal and interpersonal variables both increase and are made worse by burnout.

A second limitation is that our response rate was low and sample size was relatively small, and we may have been underpowered to detect smaller, but still relevant, associations between burnout, compassion satisfaction, and our independent variables. These data were collected from the subset of coordinators who chose to participate in the clinical trials program, and it is possible that self-selection biases influenced our dataset such that the burnout and well-being of our sample is not representative of the larger population of employees. The barriers to conducting research among healthcare providers and staff are well-documented, including the survey fatigue that can be particularly problematic for long survey batteries such as the one used here [50,51,52]. Studies consistently find that survey response rates are biased based on demographic categories and hospital role, including age, gender, socio-economic status, physician vs. non-physician status, and length of employment [50,51,52]. Two recent studies found that response biases did not impact burnout rates [9,53], suggesting that the results of this study may be generalizable even with a response rate of 35%. However, it is possible that the demographic biases noted above may influence the relationship between burnout and the individual and interpersonal independent variables that were of interest here. Related, we only conducted one focus group, and so the themes are not saturated. Given the importance of CRCs in clinical trials, more research in this area is warranted to determine whether the findings of this study are replicated.

## 5. Conclusions

Overall, CRCs reported relatively moderate levels of burnout and compassion satisfaction, and they reported feeling overwhelmed from the workload. Resilience, sleep dysfunction, stress, and incivility experienced from patients/family were significant predictors of burnout, while resilience and incivility from patients/family were significant predictors of compassion satisfaction. The fact that the variables identified here explain significant variance in burnout and compassion satisfaction is an important first step toward identifying and addressing the root causes of burnout among oncology CRCs. More research in this area is vital for optimizing the interprofessional work environment for the success of clinical trials.

## Figures and Tables

**Table 1 ijerph-18-11855-t001:** Demographic characteristics (*n* = 45).

	*n*	%
Gender identity		
Male	7	15.6
Female	38	84.4
Non-binary	0	0
Other	0	0
Race		
White	15	33.3
African American/Black	19	42.2
Asian	6	13.3
Other	4	8.9
Did not answer	1	2.2
Disease team		
Breast	4	8.9
GI	2	4.4
Head and neck	4	8.9
Leukemia/lymphoma	3	6.7
Melanoma	4	8.9
Multiple myeloma	18	40.0
Phase 1	7	15.6
Radiation oncology	1	2.2
Thoracic	2	4.4

**Table 2 ijerph-18-11855-t002:** Range and mean scores for all measures. DASS = Depression, Anxiety, and Stress Scale; ProQOL = Professional Quality of Life; Std Dev = standard deviation.

Measure	Range	Mean	Std Dev
ProQOL—Compassion satisfaction	10–50	40.3	6.9
ProQOL—Burnout	10–50	21.9	5.7
DASS—Depression	0–21	2.0	3.4
DASS—Anxiety	0–21	2.7	3.6
DASS—Stress	0–21	4.7	4.1
Workplace Incivility—Physicians and hospital personnel	12–42	23.56	7.49
Workplace Incivility—CRC teammates	10–39	22.12	8.95
Workplace Incivility—Patients and Family	10–40	20.74	7.83
Sleep Disturbance	8–40	22.4	7.1
Connor–Davidson Resilience Scale	0–100	76.6	16.5

**Table 3 ijerph-18-11855-t003:** Results of backward elimination linear regression evaluating variance in burnout for Models 1 (individual variables), 2 (interpersonal variables), and 3 (combined). Inc. = incivility.

DV: Burnout		Unstandardized		Stand.			95% CI	
Model	Predictor Variable	B	SE B	Beta	t	*p*	Lower	Upper
Reduced Personal	(Constant)	24.06	4.04		5.96	0.000	15.88	32.25
	Resilience	−0.13	0.04	−0.37	−3.30	0.002	−0.21	−0.05
	Sleep	0.25	0.09	0.32	2.68	0.011	0.06	0.44
	Stress	0.45	0.17	0.33	2.66	0.012	0.11	0.79
Reduced Interpersonal	(Constant)	15.32	2.31		6.65	0.000	10.66	19.98
	Inc. from patients, fam	0.31	0.11	0.43	2.97	0.005	0.10	0.53
Combined	(Constant)	18.43	4.42		4.17	0.000	9.47	27.40
	Resilience	−0.11	0.04	−0.32	−2.91	0.006	−0.19	−0.03
	Sleep	0.29	0.09	0.36	3.18	0.003	0.10	0.47
	Stress	0.36	0.16	0.26	2.20	0.034	0.03	0.68
	Inc. from patients, fam	0.19	0.08	0.26	2.47	0.018	0.03	0.34

**Table 4 ijerph-18-11855-t004:** Results of backward elimination linear regression evaluating variance in compassion satisfaction for Models 1 (individual variables), 2 (interpersonal variables), and 3 (combined).

DV: Compassion Satisfaction		Unstandardized		Stand.			95% CI	
Model	Predictor Variable	B	SE B	Beta	t	*p*	Lower	Upper
Reduced Personal	(Constant)	18.90	3.70		5.11	0.000	11.43	26.37
	Resilience	0.28	0.05	0.69	5.99	0.000	0.19	0.38
Reduced Interpersonal	(Constant)	45.43	2.70		16.80	0.000	39.96	50.90
	Inc. from coordinators	0.30	0.10	0.41	2.95	0.005	0.10	0.51
	Inc. from patients, fam	−0.56	0.12	−0.65	−4.72	0.000	−0.80	−0.32
Reduced Combined	(Constant)	27.38	4.47		6.13	0.000	18.35	36.42
	Resilience	0.24	0.05	0.60	5.45	0.000	0.15	0.34
	Inc. from patients, fam	−0.28	0.09	−0.32	−2.92	0.006	−0.47	−0.09

**Table 5 ijerph-18-11855-t005:** Summary of themes from qualitative analysis. CFIR constructs listed in parentheses under each theme. * All names are pseudonyms.

Theme	Quote
Burnout is caused by feeling overwhelmed from too many work responsibilities	“I’m always playing catch up…now…cause like there’s three patients that needs three different set of labs…and…you know list goes on and on.” [John*] “You are doing you…you are answering emails, you are entering data, you are trying to do it all at once. And you find yourself not being able to go to lunch because you’re like, “you know why I can’t go to lunch because I have too much to do. I can’t take 30 min to go grab something to eat and come back.” [Rachel]
Burnout can be exacerbated by a lack of understanding and engagement from physicians and leadership (Networks and Communications)	“Yeah. Sometimes we had the doctors who were like ego tripping and they are like, “Oh. You are coordinator. You are supposed to coordinate.” Then I am like, “You are the doctor. You are supposed to know your study. I’m here to supply support. Not know everything in and out and then tell you what to do.” I’m just…to remind you, “Hey. Do this. Do that. We might need to review this.” You know and then you want to challenge me like I am idiot. Like… First, I’m not going to just sit there and tolerate that type of abuse. But can you just like…we are…at the end of the day…we’re both human beings. The only thing separating you from me is that you have a medical degree and that’s it.” [Sara]
Nevertheless, there is a supportive work culture among CRCs based in teamwork (Organizational Culture)	“The support is amazing from your fellow colleague cause for me I came from a non-research field and I was kind of like a little bit doubtful how I’d be able to like fit in to it even though I came from a data aspect of it but when I arrived here everybody rallied around almost like, “You don’t have to worry and you have any problem come to me”. And they didn’t just say it. They put it into action. You don’t go to someone like come visit. Everybody has time for everybody. So that’s really amazing. Really makes the work far easier.” [Charles]
CRCs want more appreciation from leadership (Org Incentives and Rewards)	“I just wish the CTO would like appreciate the coordinators more. I feel like…if they showed us like…even if it’s,” Hey! We have lunch for all of us.” That would just be great…” [Sara]
Intervention and programming would be especially useful for new staff who are adjusting to working in a cancer setting (Needs and Resources)	“Because…because this is cancer. You…you know…you go into a room and you have been doing this for so long and your patients are coming from [unintelligible] city and they are discussing Hospice. It’s…it’s…it’s breaking your heart and you also see new people who don’t have the coping mechanisms of somebody who has been doing this for three or four years and [unintelligible].” [Rachel]

## Data Availability

Data are available on request from the corresponding author, upon reasonable request.

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
