# Peer review of "Incivility Is Associated with Burnout and Reduced Compassion Satisfaction: A Mixed-Method Study to Identify Causes of Burnout among Oncology Clinical Research Coordinators"

_ijerph, 2021, doi:10.3390/ijerph182211855_

Round 1
Reviewer 1 Report
Dear authors,
This article is a fascinating study about the consequences of working in services with high stress. The sample was tiny, but the authors included this in the limitations. The mixed methodology is adequate for this research, complementing the low sample.
Next, we want to make different appreciations:
- In the introduction, you only describe the burnout concept. Please, Complete information about the other concepts as stress, relicense, depression, incivility and explain. Why did you select these other concepts?.
- Please introduce the language and version of the Proqol instrument.
- Can you explain the reason why you eliminate the stress postraumátic of the proqol instrument? The reason is that the alphas Cronbach is not adequate, or you are not interested in this concept at this moment? This information clarification more the study.
- How long the intervention lasted with the focus group?
- Can you discuss why only respond 45 CRC of 130? Do you have any possible answers?
- Normally, in the qualitative table, you must indicate who answer the question. Include the anonymous code of the person. The code allows identifying if the same person answers all the time or is participating in responses.
I would appreciate it if you could include these answers in your study.
Reviewer 2 Report
This is a very well written paper that pulls together the available published data and provides a logical rationale for the study that follows. The analyses are appropriate, well described. My problem is the poor response rate, which you do recognize in your final paragraph. I would really like to know more about the non-respondents. Are there data available from personnel records on gender, length of employment, education? I am not quite sure that I can give you advice on how to deal with this issue, since it isn't clear what is possible - but the generalizability of your findings is compromised. Minimally there should be more discussion of the low response rate and the characteristics of the non-responders, as well as the implications for your conclusions.
Round 2
Reviewer 2 Report
The reference you added to Simonetti was incomplete. I think you should add at least one sentence recognizing that you had a low response rate. Am still not convinced about generalizability. One study doesn’t remove the issue.
Author Response
Thank you for noting that the Simonetti citation was incomplete, we have fixed this error in the resubmitted document. In addition, we found a second recent paper which indicates that, although non-respondents tend to vary in demographic categories, they do not have significantly different rates of burnout than respondents. We have added that reference as well; however, unfortunately, these are the only papers we can find that directly investigate the impact of response biases on burnout rates. As the reviewer notes, even if this finding holds to indicate that response rates do not impact burnout rate, a demographically biased and small response rate may still obscure important relationships. We reconfigured this section and added more discussion of the implications of the relatively low response rate and small sample size for this study.
(page 9-10): “A second limitation is that our response rate was low and sample size was relatively small, and we may have been underpowered to detect smaller, but still relevant, associations between burnout, compassion satisfaction, and our independent variables. These data were collected from the subset of coordinators who chose to participate in the clinical trials program, and it is possible that self-selection biases influenced our dataset such that the burnout and well-being of our sample is not representative of the larger population of employees. The barriers to conducting research among healthcare providers and staff are well documented, including the survey fatigue that can be particularly problematic for long survey batteries like the one used here (Ben-Nun 2008, O’Reilly-Shah 2017, Bethel, Rainbow et al. 2021). Studies consistently find that survey response rates are biased based on demographic categories and hospital role, including age, gender, socio-economic status, physician vs. non-physician status, and length of employment (Ben-Nun 2008, O’Reilly-Shah 2017, Bethel, Rainbow et al. 2021). Two recent studies found that response biases did not impact burnout rates (Shanafelt, West et al. 2019, Simonetti, Clinton et al. 2020), suggesting that the results of this study may be generalizable even with a response rate of 35%. However, it is possible that the demographic biases noted above may influence the relationship between burnout and the individual and interpersonal independent variables that were of interest here. Related, we only conducted one focus group and so the themes are not saturated. Given the importance of CRCs in clinical trials, more research in this area is warranted to determine whether the findings of this study are replicated.”
